# Chloroquine Inhibits Contraction Elicited by the Alpha-1 Adrenoceptor Agonist Phenylephrine in the Isolated Rat Aortas

**DOI:** 10.3390/ijms26104556

**Published:** 2025-05-09

**Authors:** Soo Hee Lee, Kyeong-Eon Park, Seong-Chun Kwon, Seong-Ho Ok, Seung Hyun Ahn, Gyujin Sim, Ju-Tae Sohn

**Affiliations:** 1Department of Anesthesiology and Pain Medicine, Gyeongsang National University Changwon Hospital, Changwon-si 51472, Gyeongsangnam-do, Republic of Korea; 2Department of Anesthesiology and Pain Medicine, Gyeongsang National University College of Medicine, Jinju-si 52727, Gyeongsangnam-do, Republic of Korea; 3Institute of Medical Science, Gyeongsang National University, Jinju-si 52727, Gyeongsangnam-do, Republic of Korea; 4Department of Anesthesiology and Pain Medicine, Gyeongsang National University College of Medicine, Gyeongsang National University Hospital, 15 Jinju-daero 816 Beon-gil, Jinju-si 52727, Gyeongsangnam-do, Republic of Korea; 5Department of Physiology, Institute of Clinical and Translational Research, Catholic Kwandong University, College of Medicine, Gangneung 25601, Gangwon-do, Republic of Korea; 6Department of Anesthesiology and Pain Medicine, Gyeongsang National University Hospital, 15 Jinju-daero 816 Beon-gil, Jinju-si 52727, Gyeongsangnam-do, Republic of Korea

**Keywords:** chloroquine, phenylephrine, contraction, alpha-1 adrenoceptor, myosin light-chain phosphorylation

## Abstract

Although chloroquine appears to inhibit the alpha-1 adrenoceptor, whether the chloroquine-mediated inhibition of phenylephrine-induced contraction is associated with the blockade of alpha-1 adrenoceptors remains unknown. This study examined the effect of chloroquine on contractions elicited by the alpha-1 adrenoceptor agonist phenylephrine in isolated rat aortas and determined the underlying mechanism. The effects of chloroquine and the alpha-1 adrenoceptor inhibitor prazosin on phenylephrine-elicited contractions were examined. The effects of the irreversible alpha-adrenoceptor inhibitor phenoxybenzamine followed by washout with fresh Krebs solution, as well as combined treatment with chloroquine and phenoxybenzamine followed by washout with fresh Krebs solution, on phenylephrine-induced contraction were investigated. Chloroquine and prazosin inhibited phenylephrine-induced contractions. However, pretreatment with prazosin eliminated the chloroquine-induced inhibition of contractions elicited by phenylephrine. Additionally, pretreatment with chloroquine and phenoxybenzamine followed by washout produced a higher contraction elicited by phenylephrine than pretreatment with phenoxybenzamine alone followed by washout. Although chloroquine did not affect the contraction induced by KCl in the endothelium-denuded aorta, it inhibited phenylephrine-induced protein kinase C (PKC) and myosin light-chain (MLC_20_) phosphorylation and Rho-kinase membrane translocation. These results suggest that chloroquine inhibits vasoconstriction elicited by phenylephrine via alpha-1 adrenoceptor inhibition, which is mediated by decreased MLC_20_ phosphorylation, the attenuation of PKC phosphorylation, and Rho-kinase membrane translocation.

## 1. Introduction

Chloroquine, an aminoquinoline derivative, is used to treat malaria and rheumatological diseases [1]. However, the acute toxicity of chloroquine results in peripheral vasodilation, which appears to be mediated by alpha-1 adrenergic inhibition and nitric oxide and histamine release [2], and reduces the increase in forearm vascular resistance caused by cold exposure [3]. Additionally, chloroquine induces the vasodilation of endothelium-denuded rat aortas precontracted with the alpha-1 adrenoceptor agonist phenylephrine and the relaxation of longitudinal ileal smooth muscle, which are mediated by the inhibition of calcium influx and calcium-activated potassium channels [4,5]. Chloroquine also causes vasodilation in endothelium-denuded rat aortas precontracted with the alpha-1 adrenoceptor agonists norepinephrine and KCl, which seems to be mediated by the attenuation of calcium entrance via voltage-operated calcium channels [6]. Moreover, chloroquine induces the vasodilation of pulmonary arteries pre-contracted with phenylephrine and KCl, which appears to be mediated by the attenuation of receptor-operated and voltage-dependent calcium channels and store-operated calcium channels in pulmonary artery smooth muscle cells [7].

Accordingly, previous studies have explored the mechanisms underlying chloroquine-induced vasodilation in arteries pre-contracted with phenylephrine or norepinephrine, and primarily focused on cellular signaling pathways downstream of activation by contractile alpha-1 adrenoceptor agonists [4,6,7]. However, they did not specifically examine the interaction between chloroquine and the alpha-1 adrenoceptor agonist concerning the alpha-1 adrenoceptor [4,6,7]. The alpha-1 adrenoceptor agonist, phenylephrine, increases intracellular calcium levels through calcium influx via receptor- and voltage-operated calcium channels and calcium release from the sarcoplasmic reticulum, leading to calcium-dependent vasocontraction [8]. In addition, phenylephrine induces vasoconstriction by inhibiting myosin light-chain phosphatase, which is mediated by the activation of protein kinase C and Rho-kinase, a mechanism known as calcium sensitization-mediated contraction [8]. Thus, the cellular signaling pathways that contribute to calcium-dependent and calcium-sensitization-mediated contractions are located downstream of alpha-1 adrenoceptor activation [8]. Voltage-operated and receptor-operated calcium channels contributing to calcium influx are located in cellular signaling pathways distal to the alpha-1 adrenoceptor activated by the alpha-1 adrenoceptor agonists phenylephrine or norepinephrine. Consequently, these previous studies could not rule out the inhibitory effect of chloroquine on the contraction elicited by phenylephrine via the inhibition of the alpha-1 adrenoceptor [4,6,7,8]. In other words, whether the chloroquine-mediated inhibition of phenylephrine-induced contraction is associated with the blockade of alpha-1 adrenoceptors remains unknown. Moreover, a competitive binding study using [^3^H] prazosin, a selective alpha-1 adrenoceptor inhibitor, indicated that a high concentration (10^−4^ M) of chloroquine increased the inhibitory constant (Ki) of prazosin in the liver plasma membrane, suggesting that high concentrations of chloroquine interact with the alpha-1 adrenoceptor [9]. Furthermore, a study to examine the underlying mechanism associated with the inhibitory effect of chloroquine on alpha-1 adrenoceptor-mediated contraction has not been reported. Therefore, this study aimed to examine the underlying mechanism responsible for the inhibitory effect of chloroquine on contractions elicited by the alpha-1 adrenoceptor agonist phenylephrine, with a particular focus on the alpha-1 adrenoceptor and its cellular signaling pathway associated with vasoconstriction mediated by the alpha-1 adrenoceptor.

## 2. Results

Chloroquine (10^−5^ and 3 × 10^−5^ M) inhibited phenylephrine-induced contractions in endothelium-intact aortas (Figure 1a—10^−5^ M chloroquine: *p* < 0.01 versus control at 3 × 10^−8^ and 10^−7^ M phenylephrine; 3 × 10^−5^ M chloroquine: *p* < 0.001 versus control at 3 × 10^−8^ and 10^−7^ M phenylephrine). In addition, chloroquine (10^−5^ and 3 × 10^−5^ M) inhibited phenylephrine-induced contractions in endothelium-denuded rat aortas (Figure 1b—10^−5^ M chloroquine: *p* < 0.001 versus control at 3 × 10^−9^ and 10^−8^ M phenylephrine; 3 × 10^−5^ M chloroquine: *p* < 0.001 versus control at 3 × 10^−9^ to 10^−7^ M phenylephrine). Moreover, chloroquine (10^−5^ and 3 × 10^−5^ M) inhibited phenylephrine-elicited contractions in endothelium-intact aortas with the nitric oxide synthase inhibitor N^w^-nitro-L-arginine methyl ester (L-NAME, 10^−4^ M) (Figure 1c—10^−5^ M chloroquine: *p* < 0.001 versus control at 10^−8^ M phenylephrine; 3 × 10^−5^ M chloroquine: *p* < 0.001 versus control at 10^−8^ to 10^−7^ M phenylephrine). High concentrations of chloroquine (3 × 10^−5^ M) inhibited phenylephrine-induced contraction in endothelium-denuded aortas and endothelium-intact aortas treated with L-NAME in a concentration-dependent manner (Figure 1b,c; 3 × 10^−5^ M chloroquine: *p* < 0.001 versus 10^−5^ M chloroquine at 10^−8^ and 3 × 10^−8^ M phenylephrine).

The selective alpha-1 adrenoceptor inhibitor prazosin (3 × 10^−9^ M) attenuated phenylephrine-induced contractions (Figure 2a; *p* < 0.001 versus the control at 3 × 10^−9^ to 3 × 10^−6^ M phenylephrine). However, combined treatment with prazosin (3 × 10^−9^ M) and chloroquine (3 × 10^−5^ M) had no effect on phenylephrine-induced contractions compared to prazosin (3 × 10^−9^ M) alone in endothelium-denuded aortas (Figure 2a). However, pretreatment with the irreversible alpha-adrenoceptor inhibitor phenoxybenzamine (5 × 10^−8^ M), followed by washout with fresh Krebs solution to remove phenoxybenzamine, inhibited phenylephrine-induced contraction (Figure 2b) in endothelium-denuded aortas. Combined pretreatment with chloroquine (3 × 10^−5^ M) and phenoxybenzamine (5 × 10^−8^ M), which was followed by washout with fresh Krebs solution to remove the combined drug treatment, increased phenylephrine-induced contraction compared with pretreatment with phenoxybenzamine alone, which was followed by washout with Krebs solution to remove phenoxybenzamine (Figure 2b; *p* < 0.001 versus phenoxybenzamine (PBZ) followed by washout at 10^−7^ to 10^−5^ M phenylephrine). Moreover, the combined treatment with the reversible alpha-adrenoceptor inhibitor phentolamine (10^−6^ M) and the irreversible alpha-adrenoceptor inhibitor phenoxybenzamine (5 × 10^−8^ M), which was followed by washout with fresh Krebs solution to remove the combined drug treatment (phentolamine and phenoxybenzamine), increased the contractions evoked by phenylephrine compared with the combined treatment with chloroquine (3 × 10^−5^ M) and phenoxybenzamine (5 × 10^−8^ M), which was followed by washout with Krebs solution to remove the combined drug treatment (chloroquine and phenoxybenzamine), in endothelium-denuded aortas (Figure 2b; *p* < 0.001 versus chloroquine and phenoxybenzamine (CQ and PBZ) followed by washout at 3 × 10^−9^ to 10^−5^ M phenylephrine).

Chloroquine (10^−5^ M) had no effect on 5-hydroxytryptamine-induced contraction in endothelium-denuded aortas (Figure 3a). In addition, high concentrations of chloroquine (3 × 10^−5^ M) slightly increased 5-hydoxytryptamine (3 × 10^−7^ M)-induced contraction in endothelium-denuded aortas (Figure 3a; *p* < 0.01 versus control; 95% confidence interval: 4.71 to 21.40). However, chloroquine (10^−5^ and 3 × 10^−5^ M) did not significantly alter the contractions elicited by KCl in endothelium-denuded aortas (Figure 3b).

In the endothelium-denuded rat aortas, and when not followed by washout with Krebs solution in either case, combined treatment with chloroquine (3 × 10^−5^ M) and phenoxybenzamine (5 × 10^−8^ M) inhibited phenylephrine-evoked contraction to a greater extent than chloroquine (3 × 10^−5^ M) alone (Appendix A; *p* < 0.001 versus chloroquine alone at 3 × 10^−8^ to 10^−5^ M phenylephrine).

Phenylephrine (3 × 10^−6^ M) significantly enhanced protein kinase C (PKC) phosphorylation in rat aortic vascular smooth muscle cells (Figure 4a; *p* < 0.001 compared to control). Chloroquine (3 × 10^−5^ M) alone did not significantly affect PKC phosphorylation (Figure 4a). However, the addition of chloroquine (3 × 10^−5^ M) suppressed phenylephrine-induced PKC phosphorylation (Figure 4a; *p* < 0.001 compared to phenylephrine alone). Similarly, phenylephrine (10^−6^ M) promoted the membrane translocation of Rho-kinase (ROCK-2) in these cells (Figure 4b; *p* < 0.001 compared to control), whereas chloroquine (3 × 10^−5^ M) effectively inhibited this phenylephrine-elicited Rho-kinase membrane translocation (Figure 4b; *p* < 0.05 compared to phenylephrine alone). Chloroquine (3 × 10^−5^ M) alone did not significantly alter Rho-kinase membrane translocation (Figure 4b). Moreover, while phenylephrine (10^−6^ M) elevated myosin light-chain (MLC_20_) phosphorylation in rat aortic vascular smooth muscle cells (Figure 4c; *p* < 0.001 versus control), chloroquine (3 × 10^−5^ M) alone did not significantly change MLC_20_ phosphorylation (Figure 4c), but it inhibited MLC_20_ phosphorylation elevated by phenylephrine (Figure 4c; *p* < 0.001 compared to phenylephrine).

Low concentrations of chloroquine (10^−5^ M) inhibited the simultaneous intracellular calcium level ([Ca^2+^]_i_) increase and contraction induced by only 10^−8^ M phenylephrine (Figure 5a,b; [Ca^2+^]_i_: *p* < 0.05 versus control; contraction: *p* < 0.01 versus control). Furthermore, a high concentration of chloroquine (3 × 10^−5^ M) inhibited the [Ca^2+^]_i_ increase induced by a narrow range of phenylephrine (10^−8^ and 3 × 10^−8^ M), whereas it inhibited the contraction induced by a wide range of phenylephrine (10^−8^ to 10^−5^ M) (Figure 5a,b; [Ca^2+^]_i_: *p* < 0.001 versus control at 10^−8^ and 3 × 10^−8^ M; contraction: *p* < 0.01 versus control at 10^−8^ to 10^−5^ M; Figure 6).

## 3. Discussion

The results of this study suggest that chloroquine inhibits contractions elicited by the alpha-1 adrenoceptor agonist phenylephrine by blocking the alpha-1 adrenoceptor. The key findings of this study are as follows: (1) chloroquine inhibited phenylephrine-induced contractions in a concentration-dependent manner, whereas prazosin pretreatment abolished the chloroquine-mediated inhibition of contractions elicited by phenylephrine; (2) pretreatment with chloroquine and phenoxybenzamine followed by washout with fresh Krebs solution induced greater phenylephrine-induced contractions than pretreatment with phenoxybenzamine alone followed by washout with fresh Krebs solution; (3) chloroquine inhibits phenylephrine-induced PKC and MLC_20_ phosphorylation and Rho-kinase membrane translocation; and (4) chloroquine had no effect on contractions induced by KCl.

Chloroquine suppressed contractions induced by the alpha-1 adrenoceptor agonist phenylephrine in both endothelium-intact and endothelium-denuded aortas, indicating that its inhibitory effect on phenylephrine-elicited contractions was not dependent on the endothelium. Chloroquine reportedly induces endothelial nitric oxide-dependent vasodilation in the endothelium-intact aorta pre-contracted with phenylephrine [4]. Furthermore, in the presence of L-NAME, chloroquine reduces phenylephrine-induced contractions in endothelium-intact aortas. Collectively, these findings suggest that the inhibitory effect of chloroquine on phenylephrine-induced contraction occurs within the vascular smooth muscle.

Consistent with previous reports, the selective alpha-1 adrenoceptor inhibitor prazosin inhibited contractions elicited by phenylephrine, suggesting that phenylephrine-induced contractions are mediated by the alpha-1 adrenoceptor [10,11,12,13]. Additionally, prior administration of the selective alpha-1 adrenoceptor inhibitor prazosin abolished chloroquine-induced reduction in phenylephrine-triggered contractions. Taken together, these results suggest that the inhibition of phenylephrine-elicited contraction by chloroquine is mediated by the blockade of the alpha-1 adrenoceptor in vascular smooth muscle. Therefore, an alpha-1 adrenoceptor protection experiment involving either the interaction between the reversible alpha-adrenoceptor inhibitor phentolamine and the irreversible alpha-adrenoceptor inhibitor phenoxybenzamine or the interaction between chloroquine and phenoxybenzamine, which were employed in previous similar experiments, was performed to examine whether chloroquine attenuates alpha-1 adrenoceptor-elicited contraction through partial occupation of the alpha-1 adrenoceptor [13,14,15]. Fentanyl, which inhibits norepinephrine-induced contraction via the blockade of the alpha-adrenoceptor, partially protects the alpha-adrenoceptor from occupation by the irreversible alpha-adrenoceptor inhibitor phenoxybenzamine [14]. Combined pretreatment with the highly selective alpha-2 adrenoceptor agonists (selectivity ratio of alpha-2 to alpha-1 adrenoceptor: 1620) dexmedetomidine and phenoxybenzamine, followed by washout with fresh Krebs solution to remove dexmedetomidine and phenoxybenzamine, led to greater phenylephrine-induced contraction than pretreatment with phenoxybenzamine alone followed by washout with fresh Krebs solution, suggesting that dexmedetomidine partially protects the alpha-1 adrenoceptor from phenoxybenzamine-induced alpha-1 adrenoceptor occupation [15,16]. Similar to previous reports involving alpha-1 adrenoceptor protection experiments [14,15], taken together, these results imply that chloroquine partially prevents the occupation of alpha-1 adrenoceptors by phenoxybenzamine because the order of magnitude of phenylephrine-induced contraction obtained after washing out the pretreatment drugs used in the alpha-1 adrenoceptor protection experiment with fresh Krebs solution was as follows (Figure 2b): 1 > pretreatment with phentolamine followed by phenoxybenzamine, followed by washing out with Krebs solution; 2 > pretreatment with chloroquine followed by phenoxybenzamine, followed by washing out with Krebs solution; 3 > pretreatment with phenoxybenzamine alone, followed by washing out with Krebs solution. In other words, chloroquine or phentolamine appears to partially protect the alpha-1 adrenoceptor from alpha-1 adrenoceptor occupation by the irreversible alpha-adrenoceptor inhibitor phenoxybenzamine, which leads to a relatively enhanced phenylephrine-elicited contraction obtained after washing out pre-treatment drugs with fresh Krebs solution. Similar to a previous study, taken together, these results suggest that the affinity of alpha-1 adrenoceptors is lower for chloroquine than for phentolamine [9]. Conversely, when not followed by washout with Krebs solution in either case, combined treatment with chloroquine and phenoxybenzamine decreased phenylephrine-induced contraction to a greater extent than chloroquine alone did, indicating that chloroquine and phenoxybenzamine additively inhibit the alpha-1 adrenoceptor. In addition, chloroquine (3 × 10^−5^ M) slightly increased 5-hydroxytryptamine (3 × 10^−7^ M)-induced contractions in endothelium-denuded aortas. Moreover, chloroquine (3 × 10^−5^ M) did not affect the contraction elicited by KCl via voltage-operated calcium channels in the endothelium-denuded aorta. Collectively, these results suggest that the chloroquine (3 × 10^−5^ M)-mediated inhibition of contraction elicited by phenylephrine appears to be a specific inhibition of alpha-1 adrenoceptors, which subsequently contributes to the inhibition of calcium sensitization and calcium influx via receptor-operated calcium channels. Consequently, further competitive binding studies are required to examine the effect of pretreatment with chloroquine or phentolamine on [^3^H] prazosin binding to alpha-1 adrenoceptors in vascular smooth muscle to confirm that chloroquine partially occupies alpha-1 adrenoceptors in the vascular smooth muscle.

Vascular smooth muscle contraction evoked by contractile agonists such as the alpha-1 adrenoceptor agonist phenylephrine is mediated by both calcium-dependent and calcium-sensitization mechanisms [8]. Phenylephrine, which acts on the G-protein-coupled alpha-1 adrenoceptor, increases intracellular calcium levels via calcium influx through receptor- and voltage-operated calcium channels and calcium release from the sarcoplasmic reticulum [8]. Increased calcium levels activate myosin light-chain kinases, leading to vasoconstriction through MLC_20_ phosphorylation [8]. Additionally, phenylephrine activates PKC and Rho-kinase, leading to vasoconstriction through increased MCL_20_ phosphorylation, which is mediated by the inhibition of myosin light-chain phosphatase due to increased PKC phosphorylation and Rho-kinase membrane translocation [8]. In the present study, phenylephrine enhanced Rho-kinase membrane translocation and PKC phosphorylation, indicating that phenylephrine-induced contractions were partially driven by calcium sensitization. However, as chloroquine (3 × 10^−5^ M) suppressed both Rho-kinase membrane translocation and PKC phosphorylation elicited by phenylephrine, the inhibitory effect of chloroquine (3 × 10^−5^ M) on phenylephrine-induced contraction was, in part, due to reduced calcium sensitization. In the current study, phenylephrine enhanced MLC_20_ phosphorylation, which appeared to be mediated by an increase in intracellular calcium and the inhibition of myosin light-chain phosphatase [8]. However, chloroquine inhibits MLC_20_ phosphorylation, likely through the suppression of intracellular calcium elevation, reduced PKC phosphorylation, and Rho-kinase membrane translocation [8]. Additionally, in the present study, chloroquine alone did not alter Rho-kinase membrane translocation or PKC and MLC_20_ phosphorylation, suggesting that the chloroquine-mediated inhibition of Rho-kinase membrane translocation and of PKC and MLC_20_ phosphorylation produced by phenylephrine is mediated by alpha-1 adrenoceptor inhibition because rat aortic vascular smooth muscle cells contain alpha-1 adrenoceptors [17]. To confirm the results of the tension study, further investigation is needed to determine whether the inhibitory effect of chloroquine on Rho-kinase membrane translocation and PKC and MLC_20_ phosphorylation induced by phenylephrine is reversible or long-lasting. In addition, further studies are needed to examine the effect of prazosin on the inhibitory action of chloroquine on Rho-kinase membrane translocation, as well as PKC and MLC_20_ phosphorylation induced by phenylephrine. Based on the tension study and Western blot analysis results, to identify whether chloroquine inhibits the [Ca^2+^]_i_ increase and contraction evoked by phenylephrine, the effect of chloroquine on the simultaneous [Ca^2+^]_i_ increase and contraction elicited by phenylephrine was examined. A low concentration (10^−5^ M) of chloroquine attenuated the [Ca^2+^]_i_ increase and contraction evoked by only 10^−8^ M phenylephrine, implying that low concentrations of chloroquine mainly attenuate [Ca^2+^]_i_ increase-mediated calcium-dependent contraction. In addition, the range (10^−8^ to 10^−5^ M phenylephrine) of phenylephrine-elicited contraction inhibited by high concentrations (3 × 10^−5^ M) of chloroquine was wider than that (by 10^−8^ and 3 × 10^−8^ M phenylephrine) of the phenylephrine-elicited [Ca^2+^]_i_ increase inhibited by it. Thus, these results imply that a high concentration of chloroquine mainly reduces the calcium-sensitization-mediated contractions elicited by phenylephrine. The inhibition of voltage-operated and receptor-operated calcium channels contributing to calcium influx, which has been identified in previous studies as a key mechanism underlying chloroquine-induced vasodilation in arteries pre-contracted with phenylephrine or norepinephrine, appears to downstream cellular responses derived from the chloroquine-mediated inhibition of alpha-1 adrenoceptor activation elicited by the alpha-1 adrenoceptor agonist phenylephrine [4,6,7]. In other words, as voltage-operated and receptor-operated calcium channels are part of the downstream cellular signaling pathway of alpha-1 adrenoceptor activation and chloroquine had no effect on contractions elicited through voltage-operated calcium channel activation by KCl, the inhibition of calcium influx observed in previous studies may be part of the downstream cellular signal response due to the chloroquine-mediated inhibition of alpha-1 adrenoceptor activation by phenylephrine [4,6,7,8]. Thus, to investigate the possible detailed sites (alpha-1 adrenoceptor, downstream pathway, and both) associated with this vasodilation, further studies regarding the effect of chloroquine on the contraction elicited by PKC and Rho-kinase simulants are needed.

Nonetheless, this study has some limitations. First, the results obtained from the alpha-1 adrenoceptor protection experiment suggest that chloroquine attenuates the contraction elicited by phenylephrine via partial occupation of the alpha-1 adrenoceptor in the rat aorta. However, a competitive binding study to examine the effect of chloroquine on [^3^H] prazosin binding to the alpha-1 adrenoceptor in vascular smooth muscle is needed to confirm the chloroquine-mediated inhibition of the alpha-1 adrenoceptor. Second, this study focused on the rat aorta, which functions as a conduit vessel. However, blood pressure regulation primarily occurs in smaller resistance arterioles, such as the mesenteric artery [18]. Third, phenylephrine, which acts on the alpha-1 adrenoceptor, produces diacylglycerol and inositol 1,4,5-triphosphate from phosphatidyl-inositol 4,5-bisphosphate by phospholipase C [19]. Then, diacylglycerol activates PKC, and inositol 1,4,5-triphosphate induces calcium release from the sarcoplasmic reticulum, contributing to contraction [19]. Thus, further study is needed on the effect of chloroquine on inositol 1,4,5-tirphosphate and diacylglycerol formation induced by phenylephrine to examine the detailed related pathways associated with the chloroquine-mediated inhibition of contraction evoked by phenylephrine. Despite these limitations, these results suggest that the toxic dose of chloroquine (fatal comatose plasma concentration of chloroquine: 5.82 × 10^−6^ M) partially produces vasodilation and subsequent hypotension, which is mediated via the alpha-1 adrenoceptor blockade [20]. Thus, in cases of severe hypotension caused by a toxic dose of chloroquine, alpha-1 adrenoceptor agonists such as phenylephrine or norepinephrine may be less effective or may require higher doses than in patients without chloroquine toxicity to achieve increased blood pressure.

## 4. Materials and Methods

This study protocol was approved by the Institutional Animal Care and Use Committee of the Gyeongsang National University (GNU-210302-R0026; date of approval: 2 March 2021). This study was conducted in compliance with the established guidelines for the care and use of laboratory animals.

### 4.1. Isolation of Rat Thoracic Aortas and Isometric Tension Measurement

Male Sprague Dawley rats weighing 230–280 g were procured from Koatech (Pyeongtaek, Republic of Korea) and euthanized using 100% CO_2_ in accordance with established ethical guidelines. The rat aorta was isolated and prepared as described previously to measure the isometric tension [21]. The thoracic cavity was surgically opened, and the descending thoracic aorta was carefully extracted. The excised aorta was subsequently placed in Krebs solution composed of 118 mM sodium chloride, 25 mM sodium bicarbonate, 11 mM glucose, 4.7 mM potassium chloride, 2.4 mM calcium chloride, 1.2 mM magnesium sulfate, and 1.2 mM monopotassium phosphate. The connective and adipose tissues surrounding the extracted rat aorta in the Krebs solution were carefully dissected under a microscope. Subsequently, the isolated descending thoracic aorta was cut into 2.5 mm segments. In certain samples, the endothelial layer was removed by inserting two 25-gauge needles into the aortic lumen and gently rolling the vessel back and forth. The descending thoracic aorta extracted from the rats was mounted in an organ bath containing a Grass isometric transducer (FT-03, Grass Instrument, Quincy, MA, USA). The bath temperature was maintained at 37 °C. According to a previous study, an initial resting tension of 24.5 mN was applied and maintained for 1.5 h to allow stabilization [22]. The Krebs solution was refreshed every 30 min to maintain optimal conditions. A continuous supply of a gas mixture containing 95% oxygen and 5% carbon dioxide was provided to ensure a stable pH of 7.4.

The integrity of the endothelium in rat aortas with an intact inner endothelium was determined using the following procedure [21]: Stable contractions were induced with phenylephrine (10^−7^ M), and acetylcholine (10^−5^ M) was added to the organ bath. If the aorta relaxed by more than 85% in response to acetylcholine after phenylephrine-induced contraction, it was identified as an endothelium-intact aorta. Subsequently, endothelium removal was verified by using a previously established method [21]. After achieving consistent contraction with phenylephrine (10^−8^ M), acetylcholine (10^−5^ M) was added to the organ bath. If the aorta showed less than 15% relaxation in response to acetylcholine, it was classified as endothelium-denuded. Rat aortas with an intact endothelium or without an endothelium were repeatedly rinsed to return to baseline resting tension following acetylcholine-induced vasodilation.

Thereafter, isolated aortas—with and without endothelia—were exposed to isotonic 60 mM KCl to generate contractions to assess the relative strength of the contractions induced by phenylephrine, 5-hydroxytryptamine, and KCl. Once contractions were sustained by isotonic 60 mM KCl, the aortas were thoroughly cleaned with fresh Krebs solution to return to the baseline resting tension. Subsequent experiments were conducted according to established protocols. In addition, as chloroquine induces endothelium-dependent nitric oxide-mediated vasodilation [4], experiments to investigate the effects of prazosin, chloroquine, phenoxybenzamine, and phentolamine, alone or combined, on phenylephrine-induced contraction in endothelium-denuded aortas were performed in endothelium-denuded aortas pretreated with the nitric oxide synthase inhibitor L-NAME (10^−4^ M) for 10 min before the addition of prazosin or phenylephrine, after washing out the pretreatment drugs with Krebs solution to avoid the effect of residual endothelium on the contraction evoked by phenylephrine in aortas with the endothelium removed.

### 4.2. Experimental Protocol

First, the effect of chloroquine on contractions elicited by the alpha-1 adrenoceptor agonist phenylephrine was examined in the aorta, with or without the endothelium, via pretreatment with chloroquine (10^−5^ and 3 × 10^−5^ M) for 20 min. In addition, since chloroquine induces vasodilation partially via endothelial nitric oxide release in endothelium-intact aortas precontracted with phenylephrine, the effect of chloroquine on the contraction elicited by phenylephrine in endothelium-intact aortas pretreated with 10^−4^ M L-NAME was examined [4]. All endothelium-intact aortas were pretreated with L-NAME for 10 min. Subsequently, some endothelium-intact aortas were post-treated with chloroquine (10^−5^ and 3 × 10^−5^ M) for an additional 20 min. Thereafter, phenylephrine (10^−9^ to 10^−5^ M) was progressively introduced into the organ bath to induce contraction, either with or without the presence of chloroquine.

Second, to examine whether the chloroquine-mediated inhibition of contractions elicited by phenylephrine is mediated by the alpha-1 adrenoceptor, the effects of the alpha-1 adrenoceptor inhibitor prazosin and chloroquine, alone or in combination, on phenylephrine-elicited contractions were examined. Endothelium-denuded aortas were pretreated with prazosin (3 × 10^−9^ M) for 20 min, followed by post-treatment with chloroquine (3 × 10^−5^ M) for an additional 20 min, or were treated with prazosin (3 × 10^−9^ M) alone for 40 min. Next, phenylephrine (10^−9^ to 10^−5^ M) was gradually introduced into the organ bath to produce contraction–response curves, either in the presence or absence of prazosin alone or in combination with prazosin and chloroquine.

Third, an alpha-1 adrenoceptor protection experiment, using the irreversible alpha-adrenoceptor inhibitor phenoxybenzamine and the reversible alpha-adrenoceptor inhibitor phentolamine, as described in a previous report, was performed to examine whether chloroquine partially occupied the alpha-1 adrenoceptor [14]. Endothelium-denuded aortas were treated with phenoxybenzamine (5 × 10^−8^ M) for 20 min and washed with fresh Krebs solution every 5 min for 60 min to remove phenoxybenzamine. Additionally, endothelium-denuded aortas were pretreated with chloroquine (3 × 10^−5^ M) or phentolamine (10^−6^ M) for 20 min, followed by post-treatment with phenoxybenzamine (5 × 10^−8^ M) for an additional 20 min. Thereafter, endothelium-denuded aortas treated with a combination of either chloroquine followed by phenoxybenzamine or phentolamine followed by phenoxybenzamine were washed with fresh Krebs solution every 5 min for 60 min to remove the combined drug treatment (chloroquine followed by phenoxybenzamine or phentolamine followed by phenoxybenzamine). Subsequently, phenylephrine (10^−9^ to 10^−5^ M) was incrementally introduced into the organ bath to evaluate the impact of pretreatment with phenoxybenzamine alone and the combined pretreatment with either chloroquine and phenoxybenzamine or phentolamine and phenoxybenzamine on the contraction elicited by phenylephrine.

Fourth, this study examined whether the inhibitory effect of chloroquine on phenylephrine-induced contractions is associated with the specific inhibition of the alpha-1 adrenoceptor. The effect of chloroquine on contractions induced by the 5-hydrxoytryptamine receptor agonist 5-hydroxytryptamine and the voltage-operated calcium channel stimulant KCl was examined. Endothelium-denuded aortas were pretreated with chloroquine (10^−5^ and 3 × 10^−5^ M) for 20 min. Subsequently, 5-hydroxytryptamine (10^−8^ to 10^−4^ M) or KCl (10 to 60 mM) was progressively introduced into the organ bath to generate concentration–response curves induced by 5-hydroxytryptamine or KCl, either with or without chloroquine.

Finally, the potential additive interaction between chloroquine and phenoxybenzamine in inhibiting the alpha-1 adrenoceptor was examined. Isolated endothelium-denuded rat aortas were pretreated either with chloroquine (3 × 10^−5^ M) alone for 40 min or with chloroquine (3 × 10^−5^ M) for 20 min, followed by phenoxybenzamine (5 × 10^−8^ M) for an additional 20 min. After pretreatment, phenylephrine was cumulatively added to the organ bath to induce contraction.

### 4.3. Cell Culture

Male Sprague Dawley rat thoracic aortas (7–8 weeks) were obtained using the method described in Section 4.1 for the culture of rat aortic vascular smooth cells with the alpha-1 adrenoceptor, as demonstrated in a previous report [17,21]. Small tissue fragments were prepared by cutting the excised thoracic aortas and transferring them to cell culture plates. The plates were placed in a cell culture incubator to promote the attachment of the tissue fragments to the plate surface. Once attachment was initiated, Dulbecco’s Modified Eagle’s Medium (DMEM) (Gibco Life Technologies, Grand Island, NY, USA), enriched with 20% fetal bovine serum (FBS), was carefully introduced. The cultures were then maintained without disturbance for the first 5 days of incubation. Primary cultured vascular smooth muscle cells were cultured in DMEM supplemented with 100 µg/mL streptomycin, 100 U/mL penicillin, and 10% FBS, following the method described in a previous study [21]. Cells from passages 3–5 were utilized and maintained under controlled conditions at 37 °C with 5% carbon dioxide in a humidified atmosphere. Before drug treatment, the vascular smooth muscle cells were incubated in serum-free medium for 16 h.

### 4.4. Membrane and Cytosolic Protein Separation Procedure

To assess Rho-kinase membrane translocation, membrane and cytosolic protein fractions were prepared using the Mem-PER™ Plus Membrane Protein Extraction Kit (Thermo Fisher Scientific, Waltham, MA, USA) following the manufacturer’s protocol [23]. In brief, harvested cell pellets were first treated with permeabilization buffer and gently mixed at 4 °C for 10 min. After incubation, the samples were centrifuged at 16,000× *g* for 15 min at 4 °C, and the supernatant was collected as the cytosolic fraction. The remaining pellet was then resuspended in solubilization buffer, mixed continuously at 4 °C for 30 min, and centrifuged again at 16,000× *g* for 15 min at 4 °C. The supernatant from this step was collected as the membrane fraction.

### 4.5. Western Blot

Western blotting was performed as previously described by Lee et al. [21]. The cells were cultivated in 100 mm or 150 mm dishes. Cells were exposed to phenylephrine (3 × 10^−6^ M) for 30 min, chloroquine (3 × 10^−5^ M) for 30 min followed by phenylephrine (3 × 10^−6^ M) for an additional 30 min, or chloroquine (3 × 10^−5^ M) alone for 60 min to evaluate PKC phosphorylation. Additionally, cells were exposed to phenylephrine (10^−6^ M) for 60 min, chloroquine (3 × 10^−5^ M) for 30 min followed by phenylephrine (10^−6^ M) for an additional 60 min, or chloroquine (3 × 10^−5^ M) alone for 90 min to evaluate Rho-kinase membrane translocation. Cells were treated with phenylephrine (10^−6^ M) for 5 min, chloroquine (3 × 10^−5^ M) for 30 min followed by phenylephrine (10^−6^ M) for another 5 min, or chloroquine (3 × 10^−5^ M) alone for 35 min to assess MLC_20_ phosphorylation. The cells were then washed twice with phosphate-buffered saline. Complete cell lysis was performed using radioimmunoprecipitation assay buffers from Cell Signaling Technology (Beverly, MA, USA) to obtain total cell lysates for immunoblot analysis. The lysates were then centrifuged at 20,000× *g* for 15 min at 4 °C. The supernatants, which contained 20 or 30 µg of protein, were heated for 10 min. Subsequently, the proteins were separated by 8–15% sodium dodecyl sulfate-polyacrylamide gel electrophoresis and transferred onto polyvinylidene difluoride membranes. After blocking with a 5% solution of bovine serum albumin or 5% skim milk in Tris-buffered saline containing 0.5% Tween-20 (TBST) at room temperature (23–27 °C) for 1 h, the membranes were incubated overnight at 4 °C with primary antibodies. These included anti-ROCK-2 (1:500), anti-PKC (1:200), anti-phospho-PKC (1:2000), anti-MLC_20_ (1:1000), and anti-phospho-MLC_20_ (1:1000) antibodies. After washing with TBST, the membranes were incubated for 1 h at room temperature with horseradish peroxidase-conjugated anti-rabbit or anti-mouse IgG antibodies, diluted 1:5000 in TBST containing 5% skim milk. Immune complexes were detected using the Westernbright^TM^ ECL Western blotting kit (Advansta, San Jose, CA, USA). Band density was quantified using ImageJ software (version 1.45s; National Institutes of Health, Bethesda, MD, USA). The levels of phosphorylation of PKC and MLC_20_ were normalized to their total form. Rho-kinase membrane translocation was calculated by dividing the membrane fraction by the total fraction (membrane fraction plus cytosolic fraction). β-actin was used as a loading control.

### 4.6. Simultaneous Intracellular Calcium Level ([Ca^2+^]_i_) and Tension Measurement in Fura-2-Loaded Aortic Strips

[Ca^2+^]_i_ was assessed using the fluorescent calcium indicator Fura-2, following the method outlined in a previous study [24,25]. Rat aortic muscle strips were incubated with 5 µM Fura-2 acetoxymethyl ester (fura-2/AM) and 0.02% Cremophor EL for 5–6 h at room temperature. Subsequently, the muscle strips were rinsed with a physiological salt solution at 37 °C for 20 min to eliminate any unhydrolyzed fura-2/AM and placed horizontally in a 7 mL organ bath maintained at a controlled temperature (37 °C). One end of the muscle strip was attached to a force–displacement transducer to record muscle contractions. Muscle strips were alternately illuminated at two excitation wavelengths (340 and 380 nm) at 48 Hz. Fluorescence intensity at 500 nm (F340/F380) was measured using a fluorometer (CAF-100, Jasco, Tokyo, Japan). The F340/F380 ratio was used to assess [Ca^2+^]_i_. The precise intracellular calcium concentration could not be determined because the dissociation constant of the calcium fluorescence indicator in the cytosol may differ from that obtained in vitro. Therefore, the F340/F380 ratios measured during resting conditions and after 60 mM KCl-induced contraction were considered 0% and 100%, respectively. The F340/F380 ratio and isometric contractions were captured using a PowerLab/400 system with the Chart program (MLT050; AD Instruments, Colorado Springs, CO, USA). The muscle strips were set under an initial resting tension of 29.4 mN for 30 min. Each experimental protocol used strips obtained from the same rat. Following the treatment of aortic strips with chloroquine (10^−5^, 3 × 10^−5^ M) alone for 10 min, simultaneous measurements of [Ca^2+^]_i_ and tension were obtained by gradually adding phenylephrine (10^−9^ to 10^−5^ M) to the organ bath, either with or without chloroquine.

### 4.7. Materials

Phenylephrine, chloroquine, prazosin, phenoxybenzamine, phentolamine, 5-hydroxytryptamine, KCl, and L-NAME were purchased from Sigma-Aldrich (St. Louis, MO, USA). Antibodies against PKC, phospho-PKC, MLC_20_, and phospho-MLC_20_ were sourced from Cell Signaling Technology (Beverly, MA, USA), whereas the ROCK-2 antibody was acquired from Santa Cruz Biotechnology (Santa Cruz, CA, USA). Fura-2/AM was purchased from Molecular Probes (Eugene, OR, USA). Phenoxybenzamine and phentolamine were dissolved in ethanol (final ethanol concentration at organ bath: 0.1%). Other drugs were dissolved in distilled water.

### 4.8. Statistical Analysis

A linear mixed-effects model was used to evaluate the impact of chloroquine, prazosin, phenoxybenzamine, and phentolamine, either individually or in combination, on contractions induced by phenylephrine, 5-hydroxytryptamine, and KCl (Stata version 14.2, StataCorp LLC, College Station, TX, USA) [26]. The Kolmogorov–Smirnov test was used to assess data normality. One-way analysis of variance (ANOVA) followed by Bonferroni’s test was used to evaluate the effects of phenylephrine and chloroquine, either separately or in combination, on PKC and MLC_20_ phosphorylation and Rho-kinase membrane translocation. A two-way repeated measures ANOVA, followed by Bonferroni’s test, was conducted to assess the impact of chloroquine on the simultaneous [Ca^2+^]_i_ and tension levels induced by phenylephrine. *p* < 0.05 was considered statistically significant.

## 5. Conclusions

The results of this study imply that a toxic dose of chloroquine attenuates vasoconstriction evoked by phenylephrine, which appears to be mediated by the blockade of the apha-1 adrenoceptor and the subsequent inhibition of PKC phosphorylation, Rho-kinase membrane translocation, and MLC_20_ phosphorylation. This study suggests that a toxic dose of chloroquine-mediated vasodilation contributes to the hypotension observed in patients with chloroquine toxicity.

## Figures and Tables

**Figure 1 ijms-26-04556-f001:**
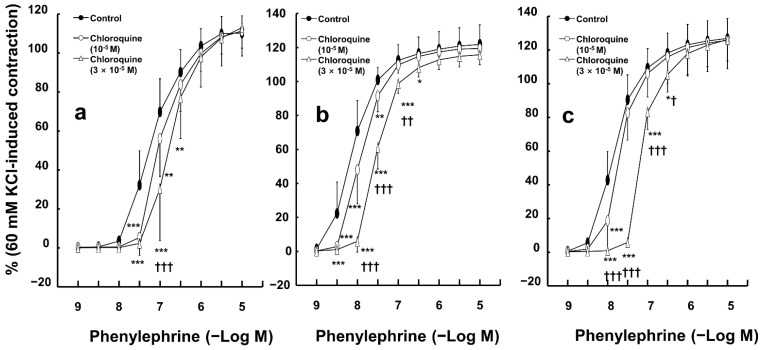
Effect of chloroquine on phenylephrine-induced contraction in endothelium-intact ((**a**), N = 6), and -denuded ((**b**), N = 6) aortas and endothelium-intact aortas pretreated with N^W^-nitro-L-arginine methyl ester (10^−4^ M, (**c**); N = 6, 6, and 5, indicating control, 10^−5^ M, and 3 × 10^−5^ M chloroquine, respectively). Data are shown as mean ± SD and expressed as the percentage of isotonic 60 mM KCl-induced contraction. N indicates the number of rats from whom isolated aortas were obtained. * *p* < 0.05, ** *p* < 0.01, and *** *p* < 0.001 versus control. † *p* < 0.05, †† *p* < 0.01, and ††† *p* < 0.001 versus 10^−5^ M chloroquine.

**Figure 2 ijms-26-04556-f002:**
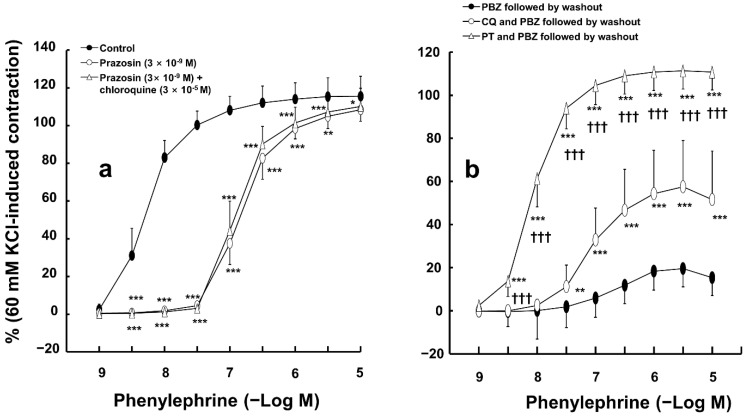
(**a**) Effect of prazosin alone and combined treatment with prazosin and chloroquine on phenylephrine-induced contraction in isolated endothelium-denuded rat aortas. Data (N = 6) are shown as mean ± SD and expressed as the percentage of isotonic 60 mM KCl-induced contraction. N indicates the number of rats from whom isolated aortas were obtained. * *p* < 0.05, ** *p* < 0.01, and *** *p* < 0.001 versus control. (**b**) Effects of pretreatment with phenoxybenzamine (PBZ, 5 × 10^−8^ M) alone followed by washout of PBZ with fresh Krebs solution (PBZ followed by washout), combined pretreatment with chloroquine (CQ, 3 × 10^−5^ M) and PBZ (5 × 10^−8^ M) followed by washout of CQ and PBZ with fresh Krebs solution (CQ and PBZ followed by washout), and combined pretreatment with phentolamine (PT, 10^−6^ M) and PBZ (5 × 10^−8^ M) followed by washout of PT and PBZ with fresh Krebs solution (PT and PBZ followed by washout) on phenylephrine-induced contraction in isolated endothelium-denuded aortas. Data (N = 10) are shown as mean ± SD and expressed as the percentage of isotonic 60 mM KCl-induced contraction. N indicates the number of rats from whom isolated aortas were obtained. ** *p* < 0.01 and *** *p* < 0.001 versus PBZ followed by washout. ††† *p* < 0.001 versus CQ and PBZ followed by washout.

**Figure 3 ijms-26-04556-f003:**
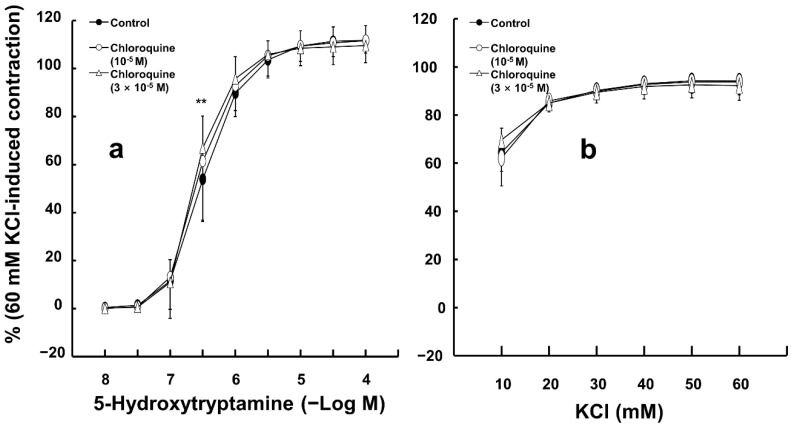
Effect of chloroquine on the contraction induced by 5-hydroxytryptamine ((**a**), N = 6) and KCl ((**b**), N = 5) in isolated endothelium-denuded rat aortas. Data are shown as mean ± SD and expressed as the percentage of isotonic 60 mM KCl-induced contraction. N indicates the number of rats from whom isolated aortas were obtained. ** *p* < 0.01 versus control.

**Figure 4 ijms-26-04556-f004:**
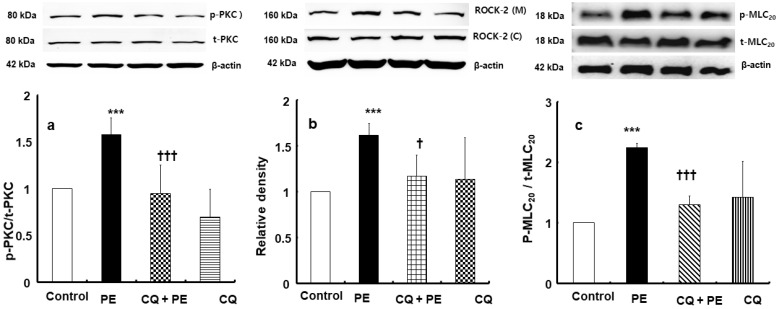
Effect of chloroquine (3 × 10^−5^ M, CQ) on phenylephrine (3 × 10^−6^ M, PE)-induced protein kinase C (PKC, (**a**), N = 5) phosphorylation, PE (10^−6^ M)-induced Rho-kinase (ROCK-2) membrane translocation ((**b**), N = 4). and PE (10^−6^ M)-induced myosin light-chain (MLC_20_, (**c**), N = 4) phosphorylation in rat aorta vascular smooth muscle cells. Data are shown as mean ± SD. N indicates the number of independent experiments. p-PKC: phosphorylated PKC; t-PKC: total PKC; ROCK-2 (M): ROCK-2 (membrane); ROCK-2 (C): ROCK-2 (cytosol); p-MLC_20_: phosphorylated MLC_20_; t-MLC_20_: total MLC_20_. *** *p* < 0.001 versus control. † *p* < 0.05 and ††† *p* < 0.001 versus PE alone.

**Figure 5 ijms-26-04556-f005:**
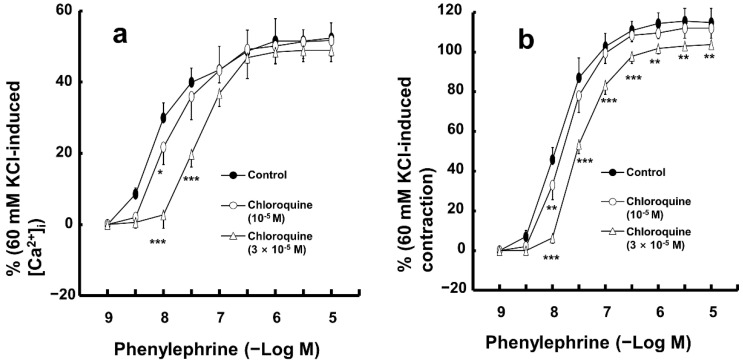
Effect of chloroquine on the simultaneous intracellular calcium level ([Ca^2+^]_i_, (**a**)) and contraction (**b**) caused by phenylephrine in endothelium-denuded aortic strip pretreated with Fura-2. [Ca^2+^]_i_ and contraction caused by phenylephrine are expressed as percentage of [Ca^2+^]_i_ and contraction caused by 60 mM KCl, respectively. Data (N = 5) are shown as mean ± SD. N indicates the number of independent experiments. * *p* < 0.05, ** *p* < 0.01, and *** *p* < 0.001 versus control.

**Figure 6 ijms-26-04556-f006:**
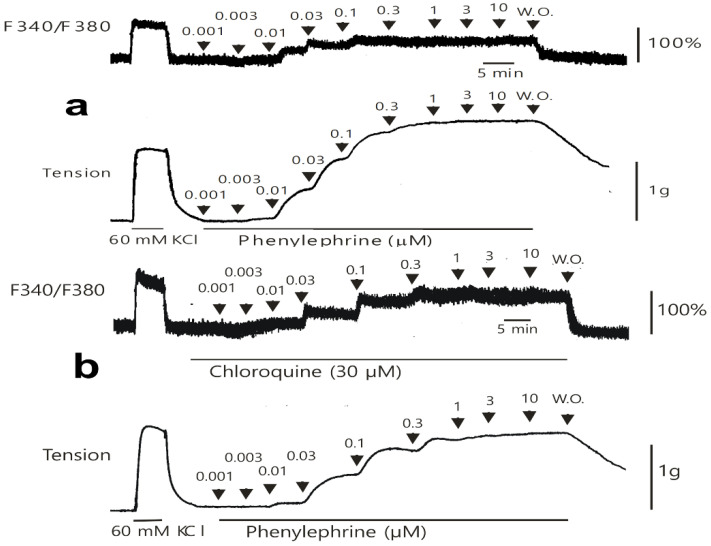
Original tracing showing the effect of chloroquine on the simultaneous intracellular calcium level (F340/F380) and contraction caused by phenylephrine in the endothelium-denuded aortic strip pretreated with Fura-2 in the absence (**a**) or presence (**b**) of chloroquine (30 μM). W.O.: washout with Krebs solution.

## Data Availability

The data presented in this study are available on reasonable request from the corresponding author.

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
