# Peer review of "Chloroquine Inhibits Contraction Elicited by the Alpha-1 Adrenoceptor Agonist Phenylephrine in the Isolated Rat Aortas"

_ijms, 2025, doi:10.3390/ijms26104556_

Round 1
Reviewer 1 Report
Comments and Suggestions for Authors
Reviewer Comments
This study showed that chloroquine inhibits phenylephrine-induced vascular contraction by antagonizing alpha-1 adrenoceptors, as evidenced by pharmacological inhibition and washout experiments. Additionally, the study confirmed that chloroquine reduces phenylephrine-stimulated activation of PKC, MLC20, Rho-kinase, and intracellular calcium in vascular smooth muscle cells (VSMCs) or rat aorta. While this study provides valuable evidence, it requires a few modifications.
Comment:
- The authors conclude that chloroquine (CQ) inhibits phenylephrine-induced vascular contraction by antagonizing alpha-1 adrenoceptors, based on washout experiments following phenoxybenzamine pretreatment. Given that isotope binding studies have demonstrated CQ's interaction with alpha-1 adrenoceptors, author should show the introduction to more clearly the study's novelty or address the specific knowledge gap.
- The authors present data on the effects of chloroquine (CQ) on 5-HT and high-K-induced contractions. Could the authors provide a clear rationale for these experiments related to alpha-1 adrenoceptors?
- Authors should describe the incubation time for each experiment (PKC phosphorylation, ROCK-2 translocation, and MLC20 phosphorylation) in isolated VSMC.
- Cells from passages 3 – 5 were utilized in your experiment. In the cultured vascular smooth muscle cells, the essential requirement is the confirmation of the presence of alpha1-adrenoceptors. Should provide experimental evidence demonstrating the expression of alpha1-adrenoceptors in VSMC preparation or relevant references for the presence of alpha1-adrenoceptors in isolated VSMC.
- In the Methods section, provide how the membrane and cytoplasmic fractions were separated from the isolated VSMCs.
- In Figure 4, the authors demonstrate that chloroquine (CQ) does not significantly alter PKC, ROCK, and MLC20 phosphorylation under basal conditions, but only CQ effect showed at PE stimulation. The author should address the potential differences for lack of CQ effect under basal conditions.
- In Figure 4A, check MW for p-PKC again? (MW 85, 82, 80, 78?)
- In Figure 5. add representative raw data tracing to enhance the clarity that CQ reduced simultaneous intracellular calcium level and contraction
- In line 334, for ethical considerations regarding animal experimentation, recommend to change 'anesthetized using 100% CO2' to 'euthanized using 100% CO2 in accordance with established ethical guidelines.'
- Cell culture line 423, specify the animal species, age, and sex used in the experiments and provide a detailed description of the isolation procedure, including the specific enzymatic or explant methods of the whole descending aorta.
- In Materials, describe how dissolved CQ, phenoxybenzamine, or phentolamine (ex, Water, DMSO, ethanol etc.)
Author Response
Responses to reviewer #1’s comments.
Thank you very much for your thoughtful comments.
- The authors conclude that chloroquine (CQ) inhibits phenylephrine-induced vascular contraction by antagonizing alpha-1 adrenoceptors, based on washout experiments following phenoxybenzamine pretreatment. Given that isotope binding studies have demonstrated CQ's interaction with alpha-1 adrenoceptors, author should show the introduction to more clearly the study's novelty or address the specific knowledge gap.
Response:
We have added the following sentences to the Introduction.
“Furthermore, a study to examine the underlying mechanism associated with the inhibitory effect of chloroquine on alpha-1 adrenoceptor-mediated contraction has not been reported.”
- The authors present data on the effects of chloroquine (CQ) on 5-HT and high-K-induced contractions. Could the authors provide a clear rationale for these experiments related to alpha-1 adrenoceptors?
Response:
We apologize for the misunderstanding. The following sentence was already present in the Experimental protocol section.
“Fourth, this study examined whether the inhibitory effect of chloroquine on phenylephrine-induced contractions is associated with the specific inhibition of the alpha-1 adrenoceptor.” Please check the highlighted text with light blue color.
- Authors should describe the incubation time for each experiment (PKC phosphorylation, ROCK-2 translocation, and MLC20 phosphorylation) in isolated VSMC.
Response:
We apologize for this misunderstanding. The incubation time was already stated in the original manuscript as follows.
“Cells were exposed to phenylephrine (3 × 10⁻⁶ M) for 30 min, chloroquine (3 × 10⁻⁵ M) for 30 min followed by phenylephrine (3 × 10⁻⁶ M) for an additional 30 min, or chloroquine (3 × 10⁻⁵ M) alone for 60 min to evaluate PKC phosphorylation. Additionally, cells were exposed to phenylephrine (10⁻⁶ M) for 60 min, chloroquine (3 × 10⁻⁵ M) for 30 min followed by phenylephrine (10⁻⁶ M) for an additional 60 min, or chloroquine (3 × 10⁻⁵ M) alone for 90 min to evaluate Rho-kinase membrane translocation. Cells were treated with phenylephrine (10⁻⁶ M) for 5 min, chloroquine (3 × 10⁻⁵ M) for 30 min followed by phenylephrine (10⁻6 M) for another 5 min, or chloroquine (3 × 10⁻⁵ M) alone for 35 min to assess MLC20 phosphorylation.”
Please check the text highlighted in light blue color.
- Cells from passages 3 – 5 were utilized in your experiment. In the cultured vascular smooth muscle cells, the essential requirement is the confirmation of the presence of alpha1-adrenoceptors. Should provide experimental evidence demonstrating the expression of alpha1-adrenoceptors in VSMC preparation or relevant references for the presence of alpha1-adrenoceptors in isolated VSMC.
Response:
Thank you very much for your positive educational comments. We have added the following sentences to the Cell culture section of the revised manuscript.
“Male Sprague-Dawley rat thoracic aortas (7–8 weeks) for culture of rat aortic vascular smooth cells with the alpha-1 adrenoceptor, as demonstrated in a previous report, were obtained using the method described in section 4.1 [17,21].”
Reference: Faber, J.E.; Yang, N.; Xin, X. Expression of alpha-adrenoceptor subtypes by smooth muscle cells and adventitial fibroblasts in rat aorta and in cell culture. J Pharmacol Exp Ther 2001, 298, 441-52.
- In the Methods section, provide how the membrane and cytoplasmic fractions were separated from the isolated VSMCs.
Response:
We have added the following section to the revised manuscript.
“4.4. Membrane and cytosolic protein separation procedure
To assess Rho-kinase membrane translocation, membrane and cytosolic protein fractions were prepared using the Mem-PER™ Plus Membrane Protein Extraction Kit (Thermo Fisher Scientific, Waltham, MA, USA) following the manufacturer’s protocol [23]. In brief, harvested cell pellets were first treated with permeabilization buffer and gently mixed at 4°C for 10 min. After incubation, the samples were centrifuged at 16,000 × g for 15 min at 4°C, and the supernatant was collected as the cytosolic fraction. The remaining pellet was then resuspended in solubilization buffer, mixed continuously at 4°C for 30 min, and centrifuged again at 16,000 × g for 15 min at 4°C. The supernatant from this step was collected as the membrane fraction.”
- In Figure 4, the authors demonstrate that chloroquine (CQ) does not significantly alter PKC, ROCK, and MLC20 phosphorylation under basal conditions, but only CQ effect showed at PE stimulation. The author should address the potential differences for lack of CQ effect under basal conditions.
Response:
We have added the following sentences to the Results and Discussion sections of the revised manuscript.
“Chloroquine (3 × 10⁻⁵ M) alone did not significantly affect PKC phosphorylation (Fig. 4a).” “Chloroquine (3 × 10⁻⁵ M) alone did not significantly alter Rho-kinase membrane translocation (Fig. 4b).”
“Chloroquine (3 × 10⁻⁵ M) alone did not significantly change MLC20 phosphorylation (Fig. 4c),”
“Additionally, in the present study, chloroquine alone did not alter Rho-kinase membrane translocation or PKC and MLC20 phosphorylation, suggesting that chloroquine-mediated inhibition of Rho-kinase membrane translocation and of PKC and MLC20 phosphorylation produced by phenylephrine is mediated by alpha-1 adrenoceptor inhibition because rat aortic vascular smooth muscle cells contain alpha-1 adrenoceptors [17].”
- In Figure 4A, check MW for p-PKC again? (MW 85, 82, 80, 78?)
Response:
Thank you very much for your important suggestion. This was an error, and we have corrected MW 85, 82, 80, 78 to MW 80 in Figure 4 in the revised manuscript.
- In Figure 5. add representative raw data tracing to enhance the clarity that CQ reduced simultaneous intracellular calcium level and contraction
Response:
We have added the original tracing showing the effect of chloroquine (3 × 10⁻⁵ M) on the simultaneous calcium-tension measurement in rat aortic strip loaded with Fura-2. Please refer to Figure 6.
- In line 334, for ethical considerations regarding animal experimentation, recommend to change 'anesthetized using 100% CO2' to 'euthanized using 100% CO2 in accordance with established ethical guidelines.'
Response:
Following your valuable advice, we have modified this sentence in the revised manuscript as follows.
“Male Sprague–Dawley rats weighing 230–280 g were procured from Koatech (Pyeongtaek, Republic of Korea) and euthanized using 100% CO2 in accordance with established ethical guidelines.”
- Cell culture line 423, specify the animal species, age, and sex used in the experiments and provide a detailed description of the isolation procedure, including the specific enzymatic or explant methods of the whole descending aorta.
Response:
We have added a detailed description regarding the cell culture to the revised manuscript as follows.
“Male Sprague-Dawley rat thoracic aortas (7–8 weeks) for culture of rat aortic vascular smooth cells with the alpha-1 adrenoceptor, as demonstrated in a previous report, were obtained using the method described in section 4.1 [17,21]. Small tissue fragments were prepared by cutting the excised thoracic aortas and transferring them to cell culture plates. The plates were placed in a cell culture incubator to promote attachment of the tissue fragments to the plate surface. Once attachment was initiated, Dulbecco's Modified Eagle's Medium (DMEM) (Gibco Life Technologies, Grand Island, NY, USA), enriched with 20% fetal bovine serum (FBS), was carefully introduced. The cultures were then maintained without disturbance for the first 5 days of incubation.”
- In Materials, describe how dissolved CQ, phenoxybenzamine, or phentolamine (ex, Water, DMSO, ethanol etc.)
Response:
We have added the following sentences to the Materials and Methods section of the revised manuscript.
“Phenoxybenzamine and phentolamine were dissolved in ethanol (final ethanol concentration at organ bath: 0.1%). Other drugs were dissolved in distilled water.”

Reviewer 2 Report
Comments and Suggestions for Authors
The current manuscript aimed to uncover the mechanistic basis of the inhibitory effect of chloroquine on phenylephrine-induced vasoconstriction, specifically focusing on alpha-1 adrenoreceptor and its signaling pathway. The manuscript reported some interesting findings regarding chloroquine effects; however, many have already been reported. The manuscript could benefit from some revision to provide a greater insights into this mechanistic basis of chloroquine on PE-induced vasoconstriction. Below are my comments.
- While comparing chloroquine's effect vs those of reversible and irreversible antagonists in washout experiments is great, there was no data of those effects prior to washout. It is not clear whether there is a synergistic or antagonistic interaction between chloroquine and these inhibitors.
- Similarly, while the study investigates the inhibition of PKC, Rho-kinase, and MLC20 phosphorylation, it doesn't look into whether chloroquine's effects are reversible and whether there is a long-term effect.
- The study lacks detailed mechanistic investigation. What's about other downstream or upstream signaling molecules besides PCK, Rho-kinase. A more detailed mechanistic study would provide insights into how chloroquine specifically interferes with alpha1 adrenergic receptor-mediated contraction at various points in the signaling cascade, which would significantly differentiate this manuscript from previously published works.
- What's about off-target effects of chloroquine?
The manuscript could benefit from a thorough revision.
Author Response
Responses to reviewer #2’s comments.
Thank you very much for your thoughtful comments.
- While comparing chloroquine's effect vs those of reversible and irreversible antagonists in washout experiments is great, there was no data of those effects prior to washout. It is not clear whether there is a synergistic or antagonistic interaction between chloroquine and these inhibitors.
Response:
We performed an additional pilot study to examine the effect of chloroquine and phenoxybenzamine, alone or combined, on the contraction induced by the alpha-1 adrenoceptor agonist phenylephrine in the endothelium-denuded aorta. The results of the pilot study showed that as pretreatment with chloroquine (3 × 10⁻⁵ M) plus phenoxybenzamine (5 × 10-8 M) inhibited phenylephrine-induced contraction to a greater extent than chloroquine (3 × 10⁻⁵ M) did alone in the endothelium-denuded rat aorta (Supplementary Figure 1), chloroquine and phenoxybenzamine appear to additively inhibit alpha-1 adrenoceptors.
Please refer to Supplementary Figure 1.
- Similarly, while the study investigates the inhibition of PKC, Rho-kinase, and MLC20 phosphorylation, it doesn't look into whether chloroquine's effects are reversible and whether there is a long-term effect.
Response:
Thank you very much for your positive comment. We have added the following sentences to the Discussion section of the revised manuscript.
“To confirm the results of the tension study, further investigation is needed to determine whether the inhibitory effect of chloroquine on Rho-kinase membrane translocation and PKC and MLC20 phosphorylation induced by phenylephrine is reversible.”
- The study lacks detailed mechanistic investigation. What's about other downstream or upstream signaling molecules besides PCK, Rho-kinase. A more detailed mechanistic study would provide insights into how chloroquine specifically interferes with alpha1 adrenergic receptor-mediated contraction at various points in the signaling cascade, which would significantly differentiate this manuscript from previously published works.
Response:
We agree with your comments. Accordingly, the following sentences have been added to the Limitations of the study in the Discussion section of the revised manuscript.
“Third, phenylephrine, which acts on the alpha-1 adrenoceptor, produces diacylglycerol and inositol 1,4,5-triphosphate from phosphatidyl-inositol 4,5-bisphosphate by phospholipase C [19]. Then, diacylglycerol activates PKC, and inositol 1,4,5-triphosphate induces calcium release from the sarcoplasmic reticulum, contributing to contraction [19]. Thus, further study is needed on the effect of chloroquine on inositol 1,4,5-tirphosphate and diacylglycerol formation induced by phenylephrine to examine the detailed related pathway associated with chloroquine-mediated inhibition of contraction evoked by phenylephrine.”
- What's about off-target effects of chloroquine?
Response:
Chloroquine is mainly used in the treatment of rheumatologic disease and malaria (Am J Emerg Med 2020: 38: 2209-17). The therapeutic margin of chloroquine is very narrow (Am J Emerg Med 2020: 38: 2209-17); therefore, chloroquine toxicity can easily occur and may produce cardiovascular (hypotension, ventricular arrhythmia, and Qt-prolongation), central nervous (headache and ataxia), respiratory (hypoxia and pulmonary edema), auditory (tinnitus and decreased acuity), ocular (diplopia and loss of visual acuity and color perception), gastrointestinal (diarrhea and vomiting), musculoskeletal (myopathy), and immunological (hypersensitivity reaction) symptoms (Am J Emerg Med 2020: 38: 2209-17).

Round 2
Reviewer 1 Report
Comments and Suggestions for Authors
The authors thoroughly addressed all raised concerns through detailed, point-by-point responses.
Author Response
Responses to Reviewer #1 comment
Thank you very much for thoughtful comments.